# Phytochemical Profiling, Antioxidant and Anti-Inflammatory Activity of Plants Belonging to the *Lavandula* Genus

**DOI:** 10.3390/molecules28010256

**Published:** 2022-12-28

**Authors:** Natalia Dobros, Katarzyna Dorota Zawada, Katarzyna Paradowska

**Affiliations:** Department of Organic and Physical Chemistry, Faculty of Pharmacy, Medical University of Warsaw, Banacha, Str. 1, 02-097 Warsaw, Poland

**Keywords:** lavender, chemical composition, antioxidant activity, anti-inflammatory activity, traditional and modern methods of extraction

## Abstract

Lavender is a valuable medicinal plant belonging to the *Lamiaceae* family. Currently 39 species are known, but only *Lavandula angustifolia* is a pharmacopoeial raw material. Lavender has a long history of medicinal use and mainly exhibits antioxidant, anti-inflammatory, sedative, antidepressant, spasmolytic, anticholinesterases, antifungal and antibacterial properties. Used internally, it relieves symptoms of mental stress and insomnia and supports digestion. Topical use of lavender in aromatherapy, neuralgia and antiseptics is also known. The constant interest in lavender, and in particular in *Lavandula angustifolia*, in the field of medicine and pharmacy is evidenced by the growing number of publications. In view of so many studies, it seems important to review traditional and modern extraction techniques that determine the chemical composition responsible for the antioxidant and anti-inflammatory effects of various extracts from the species of the *Lavandula* genus.

## 1. Introduction

Lavender is a valuable medicinal plant belonging to the *Lamiaceae* family. It is native to the Mediterranean region and grows in natural sites of the lower parts of the mountains. Lavender is cultivated as an ornamental plant in many countries in Europe, north Africa, southwest Asia, western Iran, eastern India, China and Japan (Figure 1). The *Lavandula* genus includes 39 species, but only *Lavandula angustifolia* is considered a pharmacopoeial raw material [1,2]. Lavender has a long history of medicinal use. In traditional medicine it is a popular herb used to treat multiple diseases. Lavender has antioxidant [3,4], anti-inflammatory [5,6,7,8], sedative [9], antidepressant [10], spasmolytic, anticholinesterases [11], antifungal [8] and antibacterial [1] properties. Lavender is known as a medicinal product used internally to relieve symptoms of mental stress, insomnia and digestive disorders, and externally in aromatherapy, neuralgia and as an antiseptic. Lavender decoctions and hydrolates are applied as compresses that have a beneficial effect on the skin. Lavender infusions and lavender oil in the form of inhalation have sedative and anxiolytic effects that have been confirmed in both animal and human studies [9,10,12]. Lavender is a very popular aromatic plant and is commonly used in food and cosmetics thanks to its antibacterial, antifungal, antioxidant and anti-inflammatory properties. Lavender essential oil is present in eau de toilette, lotions, soaps, shampoos and household cleaners [13,14]. The essential oil of lavender can be added to cosmetics without the need for preservatives [1]. Because of its unique composition, pro-health benefits and attractive sensory attributes lavender can be used in the processing industry as a component of products with functional properties [3]. Owing to its medicinal activities, lavender, in particular *Lavandula angustifolia*, enjoys constant interest in the medical and pharmaceutical areas, as evidenced by an increasing number of publications in the last 25 years (Figure 2).

In this review are presented the traditional and modern extraction techniques, chemical composition as well as antioxidant and anti-inflammatory activity of the extracts from different species of the *Lavandula* genus. The specific characteristics of each species are presented in Table 1 [15,16].

Recent reviews on lavender concerned mainly the anti-anxiety, antidepressant and wound-healing properties of essential oil [9,10,12,17,18]. However, to the best of our knowledge this work is the first attempt to review the data concerning the extraction and profile of other bioactive components of lavender, namely phenolic acids and flavonoids, as well as the antioxidant and anti-inflammatory properties of lavender phenolic-rich extracts. 

The species of *Lavandula* presented in Table 1 differ in terms of the height of the shrub, color of leaves, flowers and flowering period. *Lavandula coronopifolia,* which occurs naturally in northern Africa, the Arabian Peninsula and Western Asia, starts flowering the earliest, in January, whereas *Lavandula pubescens*, which occurs naturally in Syria, Jordan, Israel, Egypt, Yemen and Saudi Arabia, starts flowering from August to September. In recent years, more and more species have been cultivated outside their natural habitat. The same species grown in different areas may show morphological differences in the color of leaves, flowers, plant density or seed weight. The morphological variability, and thus the quantitative composition of individual compounds, is due to climatic conditions such as temperature, insolation, rainfall, humidity and altitude as well as environmental conditions such as fertilization, soil type and its pH level [16,19].

## 2. Extraction Process

Extraction of plant material is a process of separating bioactive compounds from the sample by means of selective solvents and standard extraction procedures [20,21]. A high extraction yield results from the appropriate selection of the solvent which should be suited to the nature of the compounds to be extracted [21,22]. The polarity of the targeted compounds is especially important. For the extraction of non-polar compounds, hexane and chloroform are used [23]. Moreover, methanol, ethanol and acetone as well as hydroalcoholic mixtures are the most generally used solvents for the extraction of polar compounds [24]. Phenolic compounds are more stable at low pH, hence the acidified hydroalcoholic solvents are frequently used for their extraction [25]. The extraction process is influenced by the physicochemical parameters of the solvents (boiling point, viscosity, density, vapor pressure and solvent power), their cost, non-flammability and non-toxicity [21,24], as well as sample preparation (drying, grinding and sample particle size) and extraction parameters (extraction time, temperature, number of extraction steps, ratio of solvent to sample and use of co-solvent) [21,22,26,27].

### 2.1. Traditional Extraction Techniques

The conventional methods include solid-liquid extraction, such as maceration, digestion, percolation, infusion, decoction and Soxhlet extraction [28]. They are the general techniques used for the extraction of medicinal plants and are mostly applied for galenical preparations. One of these is tincture which is made as a result of maceration or percolation of plant material with ethanol of a suitable concentration [2]. Maceration is based on soaking plant material in a solvent at room temperature for several hours up to several days. The process of maceration that takes place at elevated temperature is called digestion [29]. The use of repeated maceration, grinding of plant material, high temperature and stirring during the extraction process increases its efficiency [29,30]. Moreover, percolation is a continuous extraction method in which after 24 h maceration a fresh solvent flows through the comminuted plant material and thus allows it to be completely etched. Percolation usually takes less time than maceration and requires percolators, that is, vessels with a conical shape that facilitate the removal of the extracted raw material [28,29]. Infusion is obtained by macerating the ground plant material with cold or boiling water for 5 to 15 min. Decoction differs from infusion in that the raw material is boiled in parallel with water from 15 to 30 min. In both cases the thus-obtained extract is cooled and filtered [28,29]. One of the most widely used traditional methods for the extraction of heat-stable compounds of medicinal plants is Soxhlet extraction. It is a form of continuous hot extraction in which the target compounds are extracted from solids with repeated washing with organic solvents such as ethyl acetate or hexane [24]. This technique is often used in the industry, but it should be remembered that it is not environmentally friendly because it uses large amounts of toxic solvents. During the extraction of lavender essential oil with hexane, other substances such as waxes, pigments and albuminous materials are extracted in addition to volatile compounds. The hexane extracts obtained in this way can be purified, but this is time-consuming and contributes to yield loss [3]. Additionally, long-term extraction time and high solvent temperature may result in the decomposition of valuable substances [22]. The disadvantages of traditional extraction techniques are that they require long analytical times and large quantities of solvents [21,25,27,30], and also they may contribute to the degradation of thermolabile compounds [22,27], thus resulting in a lower extraction yield [21,27]. Moreover, these methods are characterized by low selectivity and reproducibility [25]. Despite numerous drawbacks, these methods are still used because they are simple, easy to implement and do not require specialized equipment [31].

### 2.2. Modern Extraction Techniques

Several modern extraction methods which are environmentally friendly and thus called green techniques, such as ultrasound-assisted extraction (UAE), microwave-assisted extraction (MAE), accelerated solvent extraction (ASE) and supercritical fluid extraction (SFE), have recently been introduced for the extraction of compounds from plant materials [27,30]. These techniques are characterized by lower volumes of organic solvents (up to 100 mL), shorter extraction times (up to 1 h, as compared to even days for, e.g., maceration) and thus lower energy consumption [24]. By using these methods better reproducibility and selectivity as well as higher quality of the extract can be obtained than with traditional techniques [21,24,25]. 

Ultrasound is sound waves with frequencies from 20 kHz to 100 MHz [30]. The best extraction frequency for lavender is 20–40 kHz [17,32,33,34]. The propagation of ultrasound waves in the liquid is related to changes in the acoustic pressure causing the cavitation phenomenon. Furthermore, cavitation leads to the formation and collapse of gas bubbles, thus resulting in mechanical disruption of the cell walls thereby causing the release of target compounds from plant material [21,35]. Ultrasound extraction is a simple extraction technique requiring only a water bath or an ultrasonic probe applied directly to the sample [30,36]. The application of ultrasound can be an alternative technique for the extraction of phenolic compounds due to the reduction in the extraction time and the amount of the solvent [22]. This technique enables the extraction of several samples at the same time in water bath; however, after the extraction process, filtration or decantation is required. The extraction efficiency is influenced by many factors such as sample size, temperature, frequency and sonication time, as well as the kind of solvent and its pH [24]. For instance, polyphenolic compounds show higher stability at a lower pH of the solvent [25].

Microwave radiation is electromagnetic radiation in the range of 300 MHz to 300 GHz [30]. Microwave power from 230 to 500 W is most often used to extract lavender [33,37,38]. Microwave-assisted extraction (MAE) can be applied both for polar and non-polar solvents; however due to the high dielectric constant, polar solvents are more often used [21]. Solvents absorb microwave radiation and are heated to a temperature above the boiling point, which results in the rapid isolation of target compounds and high extraction efficiency [22]. Generally speaking, microwave-assisted extraction could reduce extraction time and solvent consumption and also increase the purity of the obtained extract in comparison with conventional methods [22,30,36]. Additionally, this technique allows processing of several samples at the same time [39]. Extraction conditions, such as temperature and extraction time, are very important but they should be chosen with caution to avoid thermal degradation of phenolic compounds [40].

Accelerated solvent extraction (ASE), also called pressurized liquid extraction (PLE), is performed at an elevated pressure, thus resulting in a higher extraction yield in comparison with conventional techniques. The extraction efficiency is influenced by parameters such as temperature, pressure, extraction time, the nature and volume of the solvent used as well as the solid-to-liquid ratio [35,41]. The extraction is carried out at temperatures between 40 °C and 200 °C and pressures between 3.3 and 20.3 MPa, and takes up to 20 min [22,24]. Increased pressure allows one to keep the solvent in the liquid phase above its boiling point whereas high temperature accelerates the kinetics of extraction [22,29]. This leads to a reduction in surface tension and solvent viscosity, and thus to an increase in the solubility of the compounds [27]. In accelerated solvent extraction ethanol or water are used to extract polyphenols [35]. Accelerated solvent extraction can be an alternative to other techniques because of the reduction in solvent quantities and extraction time, automation of the process and the possibility of extracting samples with high humidity [24,30]. On the other hand, the use of high temperatures may contribute to the decomposition of thermolabile compounds [29]. 

Supercritical fluid extraction (SFE) is an attractive technique for the extraction of bioactive compounds present in medicinal plants. Supercritical fluids exhibit the characters of both a liquid and a gas at their critical point [40]. Their densities are similar to the density of a liquid whereas low viscosity and surface tension make them similar to a gas [29,30]. Owing to their high diffusivity, supercritical fluids increase the extraction rate. The extraction procedure usually consists of static and dynamic phases. In the first phase the vessel with sample is filled with supercritical gas and thermostated for 10–15 min. During the dynamic phase the supercritical fluid is continuously passed through the sample for about an hour and then the extract is collected in a receiver [3,42]. Solvents such as carbon dioxide, water, ethane, propane and dimethyl ether could be used in supercritical extraction [43,44]. However, the most commonly used supercritical fluid is carbon dioxide, which is considered safe for humans and the environment due to its non-toxicity, non-flammability and non-explosiveness. It enables the recovery of thermolabile compounds owing to its low critical temperature (31.1 °C) and pressure (7.38 MPa) [43,45]. Carbon dioxide is mainly used to extract carotenoids, lipids and essential oils due to its non-polarity [22,30]. The addition of up to 15–20% of co-solvent (ethanol, methanol, water) to the supercritical fluid extraction process enables the extraction of polyphenolic compounds, which are polar [46]. The total use of carbon dioxide during extraction depends on its pressure, flow rate, temperature, sample size and extraction time. In the study of Woźniak et al., 2017 [42] the CO_2_ consumption was 120–130 g for a 10 g sample and a CO_2_ flow of 1.8 g/min. The major advantage of this method is the possibility to modify the solubility of individual compounds by changing the extraction parameters such as temperature and pressure [44]. Optimal extraction conditions to obtain the maximum yield and polyphenol content from lavender flowers were a temperature of 54.5 °C, a pressure of 29.79 MPa and an extraction time of 45 min [47]. Further advantages of supercritical fluid extraction are the lack of oxygen which prevents unfavorable oxidation processes [30] and the ease of separating the extractant from the product [22,45]. Various methods used for the extraction of secondary plant metabolites are presented in Table 2.

As can be seen, in the case of lavender the majority of studies used traditional methods, i.e., maceration, infusion or digestion with water, alcohol or hydroalcoholic mixtures (17 studies). Among them the best results in terms of total yield were obtained for maceration [11,48]. As for the modern techniques, the most often used (five studies) was ultrasound-assisted extraction. This is probably due to the relatively easy access to ultrasound devices in scientific laboratories. From the analysis of various approaches to the extraction of bioactive compounds other than the essential oil components from lavender, the best method of extraction seems to be ultrasound-assisted extraction with a hydroalcoholic solvent, which decreases the extraction time compared with traditional methods. As the phenolics in lavender are not very thermolabile, an elevated temperature (up to 60 °C) can be used, increasing the yield. Unfortunately, in many cases the yield of extraction could be determined because no dry extract was obtained [37,49,50,51,52], and in some of the papers the mass of the extract and the extraction yield were not given.

**Table 2 molecules-28-00256-t002:** Extraction techniques for various lavender species.

Extraction Method	Extraction Procedure and Conditions	Yield [%]	Lavender Species	Part of Plant	Detection Method	Antioxidant Assays		Ref.
						Types of Assays	Activity	
Refluxed extraction	3 g were refluxed with 100 mL of MeOH in a water bath for 1 h. Plant material was re-extracted twice with the same solvent (2 × 100 mL). Then, the SPE procedure was used to obtain the phenolic acid fractions.	ND	*L. angustifolia*	flowers	SPE/RP-HPLC	ND		[53]
Methanol extraction	0.34 g was extracted with 30 mL MeOH.	ND	*L. stoechas*	flowers	Folin–Ciocalteu (760 nm) HPLC/ESI-MS	DPPH [mg/mL]	7.05	[51]
Ethanol extraction	300 g were extracted with 90% EtOH. Then, the dried extract was suspended in water and fractionated with ethyl acetate.	10	*L. coronopifolia*	aerial parts	UPLC- ESI- MS/MS	DPPH [µg/mL]	17.8 ± 0.8	[54]
	2 g were extracted with 10 mL of 96% EtOH for 24 h in a water bath at 45 °C.	ND	*L. angustifolia*	flowers	RP-HPLC	ND		[1]
Aqueous-ethanol extraction	2 g were extracted with 90 mL of 50% EtOH at 85 °C for 1 h.	ND	*L. angustifolia* ssp. *angustifolia*	flowers	Folin–Ciocalteu (765 nm)	DPPH [µg/mL] Fe^2^+ chelation assay [µg/mL]	95.60 ± 1.70 54.46 ± 0.55 101.40 ± 0.90 50.60 ± 0.40	[4]
			*L. angustifolia* ssp. *angustifolia* ‘Munstead’					
			*L. angustifolia. angustifolia ‘*Hidicote Blue’				96.53 ± 1.45 49.93 ± 0.75	
			*L. angustifolia* ssp. *pyrenaica*				110.36 ± 1.40 81.90 ± 1.40	
			*L. hybrida*				73.53 ± 1.25 49.90 ± 0.90	
Aqueous extraction	0.34 g was extracted with 30 mL of H_2_O	ND	*L. stoechas*	flowers	Folin–Ciocalteu (760 nm)	DPPH [mg/mL]	1.78	[51]
Infusion	1 g was extracted with 200 mL of boiling water for 10 min.	ND	*L. angustifolia*	flowers	Folin–Ciocalteu (760 nm)	ABTS [mM]	0.72 ± 0.07	[50]
	2 g was extracted with 200 mL of boiling distilled water and left to stand at room temperature for 5 min.	ND	*L. pedunculata*	flowering stems with inflorescences	HPLC-DAD-ESI/MSn	DPPH [μg/mL] TBARS [μg/mL] reducing power [μg/mL]	68 ± 0.5–191 ± 2 14 ± 1–39.1 ± 0.1 51 ± 1–167 ± 1	[16]
	1 g was homogenized in 20 mL of hot water (90 °C) for 5 min.	22.5	*L. pedunculata* ssp. *lusitanica*	aerial parts	HPLC-DAD	TEAC (μmol TE/g extract) ORAC (μmol TE/g extract) TBARS [%] Fe^2+^ chelation assay [%]	866 ± 12.5 3018 ± 91.1 100 ± 0.0 48.0 ± 5.0	[48]
Infusion with stirring	20 g was extracted with 400 mL of boiling water and stirred for 15 min.	10.8	*L. stoechas*	plant material from local market	Folin–Ciocalteu (760 nm)	DPPH [%] Fe^2+^ chelation assay superoxide anion	45 ± 0.0 84 ± 0.0 78 ± 0.0	[55]
						[%]		
Stirring	1 g was extracted with 25 mL of EtOH:H_2_O (80:20 *v*/*v*) and stirred for 1 h at 25 °C at 150 rpm.	ND	*L. pedunculata*	flowering stems with inflorescences	HPLC-DAD-ESI/MSn	DPPH [μg/mL]	87 ± 2–257 ± 7 17 ± 1–63.5 ± 0.1 67 ± 1–216 ± 6	[16]
						TBARS [μg/mL] reducing power [μg/mL]		
	30 g was extracted with 1500 mL of deionized water, heated to a specific temperature (40, 60, 80 °C ± 0.5 °C) and stirred for 90 min at 500 rpm.	0.24	*L. x hybrida*	plant material from herbal store	Folin–Ciocalteu (760 nm)	ABTS [mol Trolox/g DM]	0.216 ± 0.038	[56]
Shaking	Samples were extracted with 80% aqueous methanol and shaken at room temperature for 15 h.	ND	*L. angustifolia* ‘Lady’ *L. angustifolia* ‘Hidcote’ *L. latifolia*	leaves	HPLC-MS, Folin–Ciocalteu (735 nm)	DPPH [μmol TEAC/g DW]	14.17 ± 9.09 9.00 ± 3.00 6.56 ± 1.13	[49]
Shaking	Different protocols:	ND						
	ST1: SLE using H_2_O, shaking for 5 h							
	ST2: H_2_O:EtOH (1:1; *v/v*), shaking for 2 h							
	ST3: H_2_O:EtOH (1:1; *v/v*), shaking for 5 h		*L. spica*	plant material from local herbal market	SLE-SPE-UHPLC-MS/MS	ND		[57]
	ST4: EtOH, shaking for 5 h							
	ST5: H_2_O:MeOH (1:1; *v/v*), shaking for 2 h							
	ST6: H_2_O:MeOH (1:1; *v/v*), shaking for 2 h twice							
	ST7: H_2_O:MeOH (1:1; *v/v*), shaking for 5 h							
	ST8: MeOH, shaking for 5 h							
	2 g was extracted with 20 mL of MilliQ water and shaken for 1 h at ambient temperature	ND	*L. angustifolia*	herb	UHPLC-DA	ABTS [mmol/100 g DW]	22.00 ± 0.00 20.19 ± 2.55	[58]
Randall Extraction	2 g was extracted with 20 mL of MilliQ water by Randall extraction for 1 h at 100 °C.							
	Plant material was extracted with hexane and then with ethanol at room temperature for 48 h with plant material: solvent ratio of 1:10 (*w/w*).	12.2	*L. stoechas* *ssp. luisieri*	herb	HPLC	DPPH [µg/mL]	30.66 ± 1.9	[59]
Maceration Maceration	10 g were soaked overnight at room temperature in 200 mL of each solvent: water (w), water: ethanol (1:1) (*w/e*), ethanol (e).	22.1 21.3 12.8	*L. viridis* L’Her	aerial parts	HPLC–DAD	ORAC (w, *w/e*, e) [μmol TE/g extract]	1502.22 ±39.95 4030.26 ±02.40 1183.95 ±90.78	[11]
						TEAC (w, *w/e*, e) [μmol TE/g extract]	670.95 ± 4.24 1149.82± 17.31 332.06 ± 2.52	
	10 g was soaked overnight at room temperature in 200 mL of: water (w), water: ethanol (1:1) (*w/e*), ethanol (e).	22.4 19.4 19.6	*L. pedunculata* ssp. *lusitanica*	aerial parts	HPLC-DAD	TEAC (w, *w/e*, e) [μmol TE/g extract] ORAC (w, *w/e*, e) [μmol TE/g extract] TBARS (w, *w/e*, e) [%] Fe^2+^ chelation assay (w, *w/e*, e) [%]	569 ± 1.99 688 ± 10.59 224 ± 6.41	[48]
							1530 ± 121 2567 ± 151 861 ± 6.00	
							96 ± 2 100 ± 0 4 ± 2	
							65.9 ± 1.27 50.1 ± 0.14 32.0 ± 0.50	
	10 g was extracted with 100 mL of 70% MeOH and shaken in a water bath at 40 °C for 5 min.	ND	*L. pubescens*	aerial parts	Folin–Ciocalteu (760 nm)	DPPH [μg/mL]		
Ultrasonic- microwave-assisted extraction (UMAE)	10 g were immersed in 100 mL of 70% MeOH. The mixture was exposed to acoustic waves at 40 °C for 5 min (ultrasonic power 50 W, frequency 40 kHz, microwave power 480 W).	ND	*L. pubescens*	aerial parts	Folin–Ciocalteu (760 nm)	DPPH [μg/mL]	24.83 19.54 22.04	[38]
Ultrasonic- homogenizer-assisted extraction	10 g was extracted with 100 mL of 70% MeOH using magnetic stirring (ultrasonic power 150 W, frequency 20 kHz, 40 °C, 5 min).							
Microwave- assisted extraction (MAE)	1 g was extracted with 15 mL of 60% and 80% methanol, ethanol and acetone at 80 °C and 500 W.	ND	*L. officinalis*	flowers	UPLC-DAD-ESI-MS/MS Folin–Ciocalteu (750 nm)	CUPRAC [mmol TR/g] DPPH [µg/mL]	0.39 ± 0.01 125 ± 4.6	[37]
Ultrasonic- assisted extraction (UAE)	30 g was extracted twice with 500 mL of 80% EtOH using an ultrasonic bath for 30 min.	14.8 14.2 10.9 23.9 20.8 14.6	*L. angustifolia*	flowers * leaves inflorescence stalks	HPTLC	DPPH * [µg/mL] TBARS * [µg/mL] Fe^2+^ chelation assay * reducing power *	11.37 ± 0.69 89.36 ± 5.00 319.21 ± 21.96 25.17 ± 0.16	[60]
			*L. intermedia ‘Budrovka’*	flowers * leaves inflorescence stalks	HPTLC	DPPH * [µg/mL] TBARS * [µg/mL] Fe^2+^ chelation assay * reducing power *	17.17 ± 0.33 116.54 ± 9.96 397.71 ± 10.26 33.78 ± 2.34	
	0.5 g was immersed in 40 mL of 62.5% MeOH. Then, 10 mL of 6 M HCl was added and the mixture was submitted to ultrasounds for 15 min and refluxed in a water bath at 90 °C for 2 h.	ND	*L. vera* *(L. angustifolia)*	leaves	RP-HPLC	ND		[52]
Ultrasonic- assisted extraction (UAE)	2 g was sonicated with 20 mL of MilliQ water for 15 min at ambient temperature.	ND	*L. angustifolia*	herb	UHPLC-PDA	ABTS [mmol/100 g DW]	10.00 ± 0.00	[58]
Pulsed ultrasound-assisted extraction (PUAE)	1 g of flower residues was extracted with 40 mL of 70% EtOH using ultrasound applied in pulsed modality (frequency 26 kHz, power 200 W, temperature < 60 °C, extraction time 10 min).	ND	*L. angustifolia* ‘Rosa’	flower residues after the distillation of essential oil	RP-HPLC Folin–Ciocalteu (760 nm)	DPPH [mg TE/g of dry waste]	107.29 ± 0.05	[34]
Accelerated solvent extraction (ASE)	5 g was mixed with washed sea sand and extracted with 30 mL of 50% MeOH at 1500 PSI and 80 °C for 10 min.	20 14	*L. dentata L. stoechas*	aerial parts	RP-HPLC-DAD-MS	DPPH [µg/mL]	71.1 ± 8.7 67.0 ± 6.5	[5]
Supercritical fluid extraction (SFE)	100 g was extracted with CO_2_ at 200–300 bar and 40–60 °C for 15–45 min, CO_2_ flow rate 10 kg/h.	ND 0.53–7.28	*L. angustifolia*	flowers	HPLC Folin–Ciocalteu (765 nm) RP-HPLC	DPPH [%]	50.55 ± 0.7 78.83 ± 1.3 ND	[47,61]
	40 g was extracted at 100–300 bar and 40–60 °C for 90 min, CO_2_ flow rate 1–3 kg/h.							
Supercritical antisolvent fractionation (SAF)	The ethanolic maceration extract was dissolved in 3% EtOH and fractionated using SAF with CO_2_ at 130 bar, CO_2_ flow rate 30 g/min.	ND	*L. stoechas* ssp. *luisieri*	herb	HPLC	DPPH [µg/mL]	16.17 ± 0.7	[59]

ND—no data. * denotes analyzes performed for flowers.

## 3. Chemical Composition

Lavender flowers (*Lavandulae* flos), harvested before the flowering period, are the medicinal raw material. The main biologically active compounds of lavender are components of essential oil, phenolic compounds, triterpenes and sterols [62]. Essential oil, for which lavender is mainly known, is present in amounts from 2% to 3%. It is obtained from the flowers by hydrodistillation or steam distillation. The essential oil consists of more than a hundred components, the main of which are linalool (from 9.3% to 68.8%) and linalyl acetate (from 1.2% to 59.4%). The quality of essential oil of lavender depends both on the high content of linalool and linalyl acetate, and on their mutual proportions [3,14]. The predominant compounds include terpenes: borneol, limonene, camphene, eucalyptol, β-ocimene, 1,8-cineol, camphor, fenchone, lavandulol acetate, lavandulol, α-terpineol, β-caryophyllene, geraniol and α-pinen as well as non-terpenoid aliphatic components: octanon, octenol, octenylacetate and octanol [13,14,17,48,62]. 

An equally important group of compounds present in lavender flowers are polyphenols. They are secondary plant metabolites with various biological properties. They are found in various parts of plants: flowers, leaves, stems, fruits and seeds [63,64,65]. So far, more than 8000 polyphenolic compounds have been identified. In terms of chemical structure, they are characterized by the presence of one or more aromatic rings in a molecule and different numbers of hydroxyl groups [63]. Polyphenols are biosynthesized through the shikimic acid pathway [66]. They can be divided into several different groups: phenolic acids, flavonoids, coumarins, stilbenes and lignans [65,66]. Most phenolic compounds are found in combination with sugars, organic acids and esters [63,64,67]. 

There are two groups of phenolic acids: derivatives of hydroxybenzoic acid and hydroxycinnamic acid [67]. The phenolic acids most common in lavender are presented in Table 3. Rosmarinic acid is the most dominantly present popular hydroxycinnamic acid of the *Lavandula* genus [1,5,11,16,48,51,54]. The other representatives of this group include cinnamic acid, hydroxyhydrocinnamic acid glucoside, caffeic acid, caffeic acid 3-glucoside, chlorogenic acid, cryptochlorogenic acid, neochlorogenic acid, caftaric acid derivative, chicoric acid, p-coumaric acid, hydro-p-coumaric acid, coumaric acid hexoside, ferulic acid, ferulic acid-4-*O*-glucoside, lithospermic acid A, rosmarinic acid, salvianolic acid B (lithospermic acid B), salvianolic acid C and G, sinapic acid and yunnaneic acid F [1,5,11,16,48,49,51,52,54,57,58,68]. The hydroxybenzoic acid derivatives are less abundant in plants than the hydroxycinnamic acid derivatives. The main representatives of this group are benzoic acid, 3-hydroxybenzoic acid, 4-hydroxybenzoic acid, vanillic acid, syringic acid, protocatechuic acid, gallic acid, homoprotocatechuic acid and homovanillic acid [1,52,57,58] (Table 3). 

The presence of phenolic compounds in plant tissues protects them against adverse environmental conditions, such as high and low temperature, ultraviolet radiation, drought and salinity, and also against attack by herbivores, insects and microorganisms [64,66,69]. Besides biotic and abiotic stresses, the geographical area [16] and environmental factors such as soil composition, mineral fertilization, rainfall or temperature exert a notable effect on the content of polyphenolic compounds [49,58,64]. On the other hand, a study by Costa et al. [11] showed that the cultivation method affects the level of phenolic compounds. In vitro cultures of *L. viridis* were characterized by a higher content of phenolic compounds than wild plants, which was caused by differences between the growing conditions. The content of polyphenolic compounds also depends on the species, cultivars and selection of parts of the plant material. In the study of Blažeković et al. [60] the extracts prepared from the inflorescence stalk were characterized by a lower content of total polyphenols (3.09% and 4.54% for *Lavandula* x *intermedia* ‘Budrovka’ and *Lavandula angustifolia*, respectively) than lavender flower extracts (6.65% and 8.46%). However, the highest content of polyphenols was found in leaf extracts (7.05% and 9.20%). Likewise, Adaszyńska-Skwirzyńska and Dzięcioł [70] obtained the highest total polyphenol content for leafy stalk extracts (4.06 and 3.71 mg GAE/g d.m. for *Lavandula angustifolia* ‘Blue River’ and *Lavandula angustifolia* ‘Ellagance Purple’ extracts, respectively), and much lower for flower extracts (1.13 and 1.12 mg GAE/g d.m.). Moreover, the study by Bajalan et al. [71] showed that population variability has a significant effect on the variation in the content of phenolic compounds. Additionally, a positive correlation was found between the content of those compounds and the content of phosphorus in the soil. 

Flavonoids, occurring mainly as glycosides, are the most abundant group of polyphenols [72]. They are composed of two aromatic rings connected by a three-carbon heterocyclic ring. Depending on the differences in the structure of the heterocyclic ring, these compounds are divided into several subgroups: flavones, isoflavones, flavonols, flavanols, flavanones, anthocyanins, coumarins and chalcones [63,67]. In this review flavonoids from different subclasses present in lavender flowers are listed in Table 4. Flavones are represented by apigenin, apigenin-O-glucoside, apigenin-O-glucuronide, apigenin hexoside, genkwanin (7-methylapigenin), isoscutellarein-O-glucuronide, luteolin, luteolin-O-glucoside, luteolin-O-glucuronide, luteolin-O-hexosyl-O-glucuronide and methylluteolin-O-glucuronide [5,11,16,48,51,68]. Among the isoflavones and flavanols present in lavender formononetin and catechin occur most often, respectively [52,57]. As concerns the flavonols, quercetin, quercetin 3-O-glucoside, rutin, myricetin, taxifolin (dihydroquercetin) and fisetin (5-desoxyquercetin) are the main representatives of this group of compounds [51,57,68]. On the other hand, the representatives of flavanones include hesperetin, hesperidin, neohesperidin, naringenin, narirutin, naringin, eriodictyol, eriodictyol-O-glucuronide, eriocitrin, pinocembrin, liquiritigenin, liquiritin and vanillin [11,16,52,57,58]. More recently, new phenolic compounds such as lavandunat, lavandufurandiol, lavandufluoren, lavandupyrones A and B and lavandudiphenyls A and B have been isolated from *Lavandula angustifolia* [73]. Lavender flowers also contain coumarin derivatives (umbelliferon, herniarine), triterpenes (ursolic acid, oleanolic acid and mictomeric acid) and sterols (cholesterol, campesterol, stigamsterol and β-sitosterol) [7,62].

## 4. Antioxidant Activity

Free radicals are formed as a result of endogenous processes, i.e., enzymatic and nonenzymatic reactions in the cells, as well as due to exogenous factors such as environmental pollution, cigarette smoke, ionizing radiation, ultraviolet radiation, industrial solvents and pesticides. Free radicals contain at least one unpaired electron on the valence shell and react readily with the molecules in their vicinity, acting as prooxidants.

According to their structure, prooxidants can be classified into radical reactive species (superoxide anion radical, hydroxyl radical, peroxyl radical, nitric oxide radical) and non-free radical reactive species (peroxynitrite, singlet oxygen, hydrogen peroxide). Their overproduction leads to imbalance in the organism and damage to lipids, proteins and DNA due to their high reactivity. The ROS and RNS thus contribute to premature skin aging and the development of multiple diseases such as diabetes mellitus, hypertension, atherosclerosis, cardiovascular disease, liver diseases, renal failure, arthritis, cancer, as well as Alzheimer’s and Parkinson’s disease [74,75,76].

The antioxidant activity of polyphenols results from their ability to prevent the formation of free radicals or scavenge the reactive oxygen species. They can donate a hydrogen atom or an electron showing reducing properties [74]. These compounds can prevent oxidation processes by inhibition of xanthine oxidase, induction of antioxidant enzymes such as superoxide dismutase, glutathione dismutase, glutathione peroxidase and catalase, and chelating capacity due to metal ions [67,75]. Besides binding ferrous and copper ions, phenolic compounds also absorb UV radiation [77]. 

The antioxidant activity of phenolic compounds is associated with the presence and position of hydroxyl groups in their molecule. Stronger antioxidant properties were found for hydroxycinnamic acids than for hydroxybenzoic acids. Reports indicate that the highest antioxidant activity is demonstrated by rosmarinic acid, followed by chicoric acid and caffeic acid [78,79]. In addition, the presence of a methoxy group increases the antioxidant activity of phenolic acids. Ferulic acid having one group attached to the benzene ring is a less effective antioxidant than synapic acid which has two methoxy groups [80]. On the other hand, the antioxidant properties of flavonols are the consequence of the presence of a hydroxyl group at the C3 position of the flavonoid skeleton [81]. 

The methods of measuring antioxidant capacity can be described according to the reaction mechanism, namely SET (single electron transfer) or HAT (hydrogen atom transfer) mechanism, or both. The SET mechanism is based on donating one electron, whereas the HAT mechanism is based on the hydrogen atom transfer by the antioxidant [77,82]. 

The SET methods based on the reduction of ions include the FRAP and CUPRAC assays. The FRAP (Ferric Reducing Antioxidant Power) assay measures the reduction of ferric ion (Fe^3+^) to ferrous ion (Fe^2+^) through the donation of an electron. This reaction is carried out in an acidic medium (pH 3.6) to maintain the solubility of iron and leads to the formation of an intensely blue ferrous-tripyridyl-S-triazine (TPTZ) complex with an absorption maximum at 593 nm. TPTZ is the most popular iron-binding ligand used in the FRAP assay. The absorbance value of the sample is directly proportional to the concentration of the antioxidant. Moreover, in the CUPRAC (Cupric Ion Reducing Antioxidant Capacity) method, the reduction of Cu(II) ions to Cu(I) ions is used to measure the antioxidant capacity. The Cu(I) ions form an orange-yellow complex with neocuproin with an absorption maximum at 450 nm [37,77]. 

The ORAC assay (Oxygen Radical Absorbance Capacity) is one of the methods based on the HAT mechanism. It uses the process of deactivating peroxide radicals. It is based on measuring the decrease in the fluorescence intensity of fluorescein—a molecular probe that is oxidized by peroxide radicals. In this assay, AAPH (2,2′-azobis (2-amidinopropane) dihydrochloride) is mostly used as a source of free radicals. The degradation of fluorescein is slower when there are more antioxidants in the sample [48,82].

The most commonly used methods based on both mechanisms are the DPPH and the ABTS assays. DPPH (2,2-diphenyl-1-picrylhydrazyl) is a stable radical that can accept an electron or a hydrogen atom [83]. The DPPH alcoholic solution is dark purple in color with an absorption maximum at 517 nm. As a result of the reaction with phenolic compounds the ethanolic solution of the DPPH radical changes its color to light yellow [77]. The decrease in absorbance of the solution or signal intensity of electron paramagnetic resonance spectroscopy (EPR) is proportional to the amount of the reduced DPPH form that was formed during the reaction [74,84]. On the other hand, the ABTS method uses the ABTS radical cation (2,2′-azobis (3-ethylbenzothiazoline-6-sulfonate) which is formed as a result of chemical or enzymatic reactions. The ABTS radicals produced during the reaction with potassium persulfate are blue-green in color and have an absorption maximum at 734 nm. In the presence of antioxidants, the radical cation is reduced, resulting in discoloration of the solution [58,79], proportional to the antioxidant content in the sample [50]. This assay enables the measurement of the antioxidant activity of hydrophilic and lipophilic compounds due to the solubility of the ABTS radical in both aqueous and organic solvents [58]. Antioxidant activity of the extracts is usually expressed as Trolox equivalents, the synthetic vitamin E derivative [48,58,79]. In addition, Relative Antioxidant Capacity Index (RACI) can be used to comprehensively determine the total antioxidant capacity of the samples as determined using various methods, including ABTS, DPPH and ORAC [85].

As a different approach, the TBARS method (Thiobarbituric Acid Reactive Substances Assay) is used to measure lipid peroxidation products. It is based on the spectrophotometric measurement of malondialdehyde (MDA) produced during lipid peroxidation. As a result of the reaction of thiobarbituric acid (TBA) with MDA, a pink complex is formed that absorbs at a wavelength of 532–535 nm. The absorbance value is proportional to the concentration of MDA, a compound that is commonly used as an oxidative stress marker. In the presence of antioxidants the formation of MDA is inhibited [82]. The results of the antioxidant activity assessment of lavender extracts are reported in Table 2. Significant differences in antioxidant activity can result from the sample origin, the extraction method used, as well as the differences between species. Unfortunately, the way in which the results of antioxidant activity are presented by different scientific teams is difficult to compare due to different conversions for extracts, different units and substances as equivalents. In addition, the extraction conditions such as different concentrations of reagents, solvents and temperature are responsible for different obtained values even for the same substance marked with the same test. We have made an effort to standardize these values; however, the publications lack many details required for such conversion. Robu et al. [4] performed a comparative study of antioxidant activity for various cultivars of *Lavandula angustifolia* with *Lavandula hybrida*. The highest result was obtained for *Lavandula hybrida* (IC_50_ = 49.90 and 73.53 μg/mL for ferrous ion chelating and DPPH assay, respectively), followed by *Lavandula angustifolia* ‘Hidcote Blue’ (IC_50_ = 49.93 and 96.53 μg/mL), and *Lavandula angustifolia* ‘Munstead’ (IC_50_ = 50.60 and 101.40 μg/mL). Whereas, in the study by Blazeković et al. [60] in most of the tests a slightly higher antioxidant activity was observed for *Lavandula angustifolia* extracts as compared with *Lavandula* x *intermedia* ‘Budrovka’ extracts, which could be due to their higher polyphenolic contents. However, in both studies all *Lavandula* extracts showed a concentration-dependent antioxidant activity, wherein the strongest DPPH-radical-scavenging and iron-chelating activity were observed at a higher concentration of the extracts. On the other hand, in the study by Ahn et al. [49] the aqueous-methanolic leaf extract of *Lavandula angustifolia* ‘Lady’ afforded an over twice as high DPPH value (14.17 μmol TEAC/g DW) than the *Lavandula latifolia* extract (6.56 μmol TEAC/g DW). In another study, the antioxidant activity of methanolic extracts of different species and cultivars of lavender was analyzed [68]. The ability to scavenge free radicals of plant extracts decreased in the following order: *Lavandula viridis* (99.47% of inhibition), *Lavandula stoechas* (95.18%), *Lavandula angustifolia* ‘Rosea’ (93.92%), *Lavandula lanata* (92.78%), *Lavandula angustifolia* ‘Afropurpurea’ (92.09%) and *Lavandula angustifolia* (91.51%). The highest inhibition of the *Lavandula viridis* extract can be due to the presence of ferulic acid which was found only in this sample.

The above studies showed a significant antiradical activity of lavender extracts. On the other hand, a study by Celik et al. [37] showed that the antioxidant activity of the microwave-assisted lavender extract is significantly lower than that of other plants of the *Lamiaceae* family. The highest TAC values evaluated by the CUPRAC assay were obtained for *Origanum majorana* (0.66 mmol TR/g) and *Mentha pulegium* (0.58 mmol TR/g) extracts whereas the lowest ones were obtained for the *Lavandula officinalis* extracts (0.39 mmol TR/g). Likewise, Nicolai et al. [86] observed that ultrasound-assisted ethanolic extracts of *Melissa officinalis*, *Salvia officinalis*, *Hypericum perforatum* and *Rosmarinum officinalis* afforded the strongest DPPH-radical-scavenging activity of 95.2%, 94.7%, 92.7% and 72.5%, respectively. In contrast, the *Lavandula angustifolia* extract had a significantly lower value equal to 17.7%.

## 5. Anti-Inflammatory Activity

Inflammation is a protective response elicited by numerous biological (bacteria, fungi, viruses, endo- and exotoxins), chemical (acids, bases, carrageenan) and physical (mechanical factors, ultrasonic waves, ionizing radiation, magnetic field) agents. Each of these factors triggers a body defensive reaction [87]. Inflammation can be acute or chronic, and each type is related to different effects. In the acute phase, lasting from several minutes to a few days, neutrophils migrate from dilated blood vessels to the site of infection, causing redness, swelling and pain. Moreover, the state of chronic inflammation can lead to the development of multiple diseases such as rheumatoid arthritis, gout, cardiovascular diseases, diabetes, bowel diseases, Alzheimer’s disease, cancer and depression.

Inflammation is associated with excessive activity of the immune system by releasing inflammatory cells such as macrophages, neutrophils and lymphocytes [88,89]. The key role of the immune system is also related to the expression of inflammatory mediators, the most important of which are vasoactive amines (histamine, serotonin) and peptide (bradykinin), arachidonic acid metabolites (eicosanoids—prostaglandins, thromboxanes, leukotrienes and lipoxins), proinflammatory cytokines (IL-1β, IL-6, IL-8, IL-12, TNF-α), chemokines and proteolytic enzymes (elastin, cathepsins, matrix metalloproteinases) [88,90]. Molecules such as nuclear factor kappa-light-chain-enhancer of activated B cells (NF-ĸB), transforming growth factor β (TGF-β), reactive oxygen species (ROS), reactive nitrogen species (RNS), inducible nitric oxide synthase (iNOS) and cyclooxygenases (COX) are also released during the inflammatory process [88,89,91].

For testing anti-inflammatory activity several animal experimental models are used (Table 5) such as the carrageenan-induced paw edema in mice [5] and in rats [92], the formalin test in mice [93], the croton-oil-induced ear edema in mice [7], the TPA-induced ear edema in mice [18], and the cell line stimulated with LPS [16]. In these studies anti-inflammatory drugs such as dexamethasone [16,93] or the nonsteroidal anti-inflammatory drugs (NSAIDs) such as indomethacin [7,93], aspirin [6], ibuprofen [18] and diclofenac [5] are used as a positive control. NSAIDs work by inhibiting cyclooxygenase, the enzyme responsible for eicosanoid synthesis [94]. Cyclooxygenase (COX) converts arachidonic acid to prostaglandin G2, then into prostaglandin H2, and finally to prostaglandins (PGF2_α_ and PGE_2_), prostacyclin (PGI_2_) and thromboxane A2 (TXA_2_) [91]. There are three cyclooxygenase isoforms. COX-1 is a constitutive form that plays a role in normal physiological processes. COX-2 is an inducible form involved in inflammation, whereas the least known COX-3 is associated with the central nervous system. The majority of NSAIDs inhibit both COX-1 and COX-2 [91,95]. On the other hand, lipoxygenase (5-LOX) metabolizes arachidonic acid to 5-hydroxyeicosatetraenoic acids and leukotrienes (LTs) [88,96]. 

Research has confirmed that the anti-inflammatory activity of lavender is due to the presence of essential oil components, non-volatile terpenoids and polyphenols. In the study by Carrascoet al. [13], thymol, fenchone and camphor from *Lavandula stoechas* essential oil showed an inhibitory effect on lipoxygenase (LOX). Luo et al. [18] observed that essential oil of *Lavandula angustifolia* reduced the expression of inflammatory mediators such as IL-6, NF-κB and TNF-α in the TPA-induced ear edema model in mice. Furthermore, in the study by Sosa et al. [7] the higher inhibition of croton-oil-induced ear edema in mice was observed for ethanolic extracts of *Lavandula multifida* than for aqueous extracts, but in both cases the antiphlogistic activity was dose-dependent. The terpenoids ursolic acid, oleanolic acid and maslinic acid and phenolic monoterpene carvacrol were obtained as a result of fractionation of the ethanolic extract. Numerous scientific reports [5,6,16,92] indicate that plants of the *Lavandula* genus are a source of polyphenolic compounds with anti-inflammatory activity. One of many antiphlogistic mechanisms of polyphenols is their ability to inhibit enzymes including iNOS, LOX and COX [92,96]. The anti-inflammatory activity of the *Lavandula* extracts was assessed using a fluorimetric test based on the detection of prostaglandin G2 generated by the COX enzyme. Cyclooxygenase COX-2 was more strongly inhibited by lavender extracts than cyclooxygenase COX-1. Shaikh et al. [6] observed that the ethanolic fraction of *Lavandula bipinnata* obtained by Soxhlet extraction inhibited COX-2 by 50%, compared to COX-1 by 19%. A high-performance thin-layer chromatography (HPTLC) analysis showed that the extract was rich in flavonoids. Likewise, Husseini et al. [93] demonstrated that the anti-inflammatory activity of hydroalcoholic macerate of *Lavandula officinalis* was associated with the inhibition of COX-2 by 45% and COX-1 by 33%. In addition, the anti-inflammatory effect of COX-2 increased with the increasing concentration of the extract. Researchers also showed that the extract inhibited the chronic (inflammatory) phase of formalin-induced pain in mice, whereas it had no effect on the acute (neurogenic) phase. Moreover, in the study of Algieri et al. [5], *Lavandula stoechas* and *Lavandula dentata* hydromethanolic extracts, obtained using accelerated solvent extraction (ASE), showed anti-inflammatory activity against carrageenan-induced paw edema in mice. The hydromethanolic extract of *Lavandula stoechas* at a dose of 100 mg/kg significantly decreased the expression of the pro-inflammatory cytokines IL-1β, IL-6 and TNF-α and the enzymes iNOS, COX-2 and MMP-9, whereas the extract of *Lavandula dentata* decreased the expression of only iNOS, COX-2 and IL-1β. The qualitative analysis of the extracts performed using RP-HPLC-DAD-MS showed that they contained the phenolic acids hydroxybenzoic acid, hydroxycinnamic acid and their derivatives as well as flavones which are a subclass of flavonoids. Similarly, Yassine et al. [92] observed the anti-inflammatory activity of the extracts of *Lavandula stoechas* on carrageenan-induced paw edema in rats. A significant inhibition of edema was found in the case of hydroethanolic extract obtained using ultrasound-assisted extraction and two fractions rich in flavonoids and mucilages, whereas that effect was not observed for the tannin fraction. Furthermore, Lopes et al. [16] used the mouse macrophage-like cell line RAW 264.7 stimulated with LPS to study the anti-inflammatory activity of the extracts of *Lavandula pedunculata*. The hydroethanolic extracts displayed a higher anti-inflammatory potential through inhibition of NO production than the aqueous extracts (infusions). 

## 6. Conclusions

According to Pharmacopoeia XI, only *Lavandula angustifolia* is currently recognized as a medicinal raw material. However, in recent years other species such as *Lavandula stoechas*, *Lavandula intermedia*, *Lavandula latifolia*, *Lavandula dentata* and *Lavandula luisieri* have also gained increasing interest in the medical and pharmaceutical areas. Many of these species are a rich source of phenolic compounds such as phenolic acids and flavonoids, thus resulting in antioxidant and anti-inflammatory properties. However, in order to obtain a high-quality extract, it is crucial to select the appropriate extraction method.

In this review are presented both the traditional and modern extraction methods, also called green techniques. These techniques are characterized by lower consumption of the solvents and shorter extraction times, which means lower energy consumption. By using these methods better reproducibility and selectivity as well as higher quality of the extracts can be obtained than by traditional techniques. Each of the presented methods has its advantages and disadvantages. Which method a given laboratory chooses depends on many factors, such as the availability of equipment, solvents and trained staff, as well as maintenance costs.

However, since in the analyzed studies there was a great variation in both extraction protocols and plant material, there is a need for a systematic study comparing various extraction techniques of the same plant material to reliably recommend the optimal approach. Further studies are needed with special attention paid to the optimization extraction and activity-guided extraction. Additionally, a key issue is to standardize the units and substances as equivalents in which the obtained results of activity determination are presented. There is also a gap concerning the extraction and analysis of lavender flower residues left after the distillation of essential oil—currently the main product of lavender—which could be a valuable source of phenolics exhibiting antioxidant and anti-inflammatory properties.

## Figures and Tables

**Figure 1 molecules-28-00256-f001:**
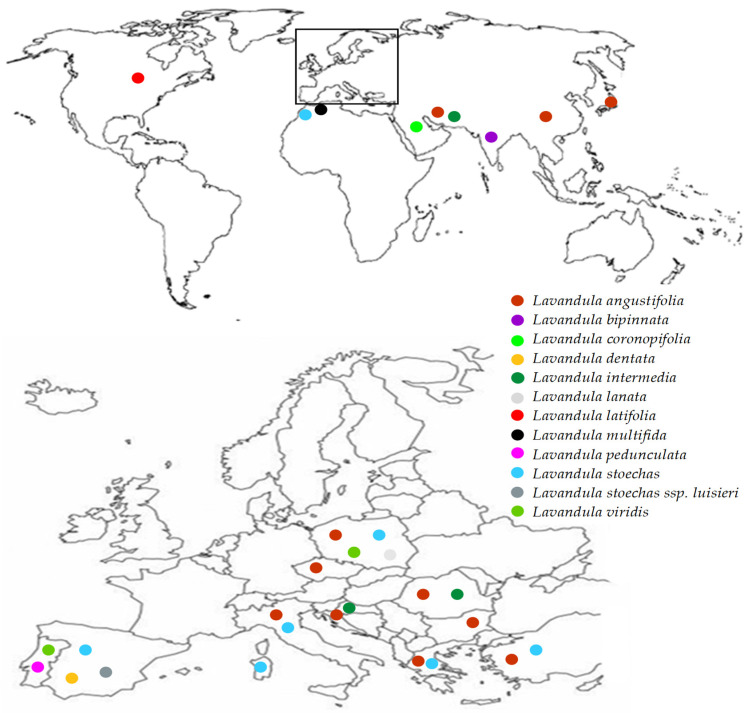
Global distribution map of various species belonging to the *Lavandula* genus. Inset: distribution of *Lavandula* in Europe.

**Figure 2 molecules-28-00256-f002:**
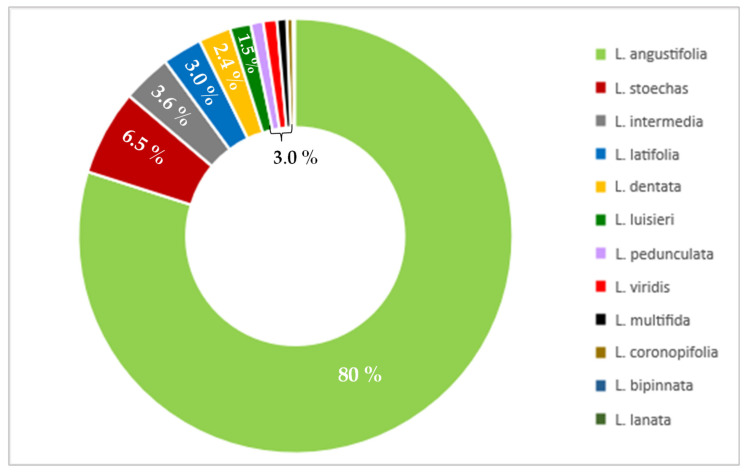
Percentage of publications for various lavender species in the last 25 years (PubMed, November 2022).

**Table 1 molecules-28-00256-t001:** The specific characteristics of different species of Lavandula [15,16].

Systematic Name	Common Name	Height of Shrub [cm]	Color of Leaves	Color of Flowers	Flowering Period	Place of Native Origin
*Lavandula angustifolia* (*L. officinalis, L. vera*)	British lavender	to 50	grey leaves becoming green as they mature	shades of blue or mauve, white	mid-June to July	Southwest and South Central Europe in mountainous areas over 1500 m
*Lavandula bipinnata*	-	15–100	green	pale blue	August	Central and South India
*Lavandula coronopifolia*	-	to 80	green to grey-green	sky blue to lilac	January to April	Cape Verde Islands, North Africa, Western Asia, Arabian Peninsula
*Lavandula dentata*	-	50–100	green to grey-green	shades of violet-blue to mauve	June to August	South Spain, Balearic Islands, North Africa, South West Arabian Peninsula, Ethiopia
*Lavandula intermedia* (*L. hybrida*)	lavandin	60–150	grey tomentose	shades of lilac-purple to white	late June to July	France, Spain, Italy
*Lavandula lanata*	woolly lavender	50–80	leaves covered with dense white woolly hairs	dark purple	mid-to late July	mountainous areas in South Spain over 2000 m
*Lavandula latifolia* (*L. spica*)	spike lavender	50–70 (100)	grey	blue to mauve	From mid-July	Southwest and South Central Europe to 1000 m
*Lavandula multifida*	Egyptian lavender	to 40	grey-green	violet to blue-violet	June to September	South Spain and Italy, North Africa
*Lavandula pedunculata*	butterfly lavender	to 60	grey-green	mauve	June to July	Iberian Peninsula, North Africa and Turkey
*Lavandula pubescens*	-	30–60	green	violet-blue	August to September	Syria, Jordan, Israel, Egypt, Saudi Arabia, Yemen
*Lavandula stoechas*	French lavender	40–70	grey tomentose	black-purple to mauve	May to September	Mediterranean basin
*Lavandula stoechas* subsp. *luisieri* (*Lavandula luisieri*)	Spanish lavender	40–60	green	mauve	spring	Southwest Spain, Central and South Portugal
*Lavandula viridis*	white lavender	30–50	green	white	spring	Southwest Spain, South Portugal, Madeira

**Table 3 molecules-28-00256-t003:** Hydroxycinnamic and hydroxybenzoic acids in different species of lavender.

Hydroxycinnamic Acids	Species	Part of Plant	Contents [μg/g]	Ref.
cinnamic acid	*Lavandula angustifolia*	herb	0.028–0.050	[58]
	*Lavandula angustifolia*	flowers	0.001	[1]
hydroxycinnamic acid glucoside	*Lavandula dentata*	aerial parts	n.q.	[5]
	*Lavandula stoechas*		n.q.	
caffeic acid (3,4-dihydroxycinnamic acid)	*Lavandula angustifolia*	herb	0.018–0.062	[58]
	*Lavandula angustifolia*	flowers	0.015	[1]
	*Lavandula angustifolia*	herb	n.q.	[68]
	*Lavandula angustifolia* ‘Rosea’		n.q.	
	*Lavandula angustifolia* ‘Afropurpurea’		n.q.	
	*Lavandula lanata*		n.q.	
	*Lavandula stoechas*		n.q.	
	*Lavandula viridis*		n.q.	
	*Lavandula coronopifolia*	aerial parts	n.q.	[54]
	*Lavandula pedunculata*	flowering stems	n.q.	[16]
	*Lavandula spica*	herb	0.585	[57]
	*Lavandula stoechas*	flowers	n.q.	[51]
	*Lavandula vera*	leaves	0.001	[52]
caffeic acid 3-glucoside	*Lavandula angustifolia* ‘Lady’	leaves	n.q.	[49]
chlorogenic acid (3-O-caffeoylquinic acid)	*Lavandula angustifolia*	flowers	0.007	[1]
	*Lavandula angustifolia*	herb	n.q.	[68]
	*Lavandula angustifolia* ‘Rosea’		n.q.	
	*Lavandula angustifolia* ‘Afropurpurea’		n.q.	
	*Lavandula lanata*		n.q.	
	*Lavandula stoechas*		n.q.	
	*Lavandula viridis*		n.q.	
	*Lavandula pedunculata* subsp. *lusitanica*	aerial parts	0.012	[48]
	*Lavandula viridis* L’Her	aerial parts	0.096	[11]
cryptochlorogenic acid (4-O-caffeoylquinic acid)	*Lavandula pedunculata* subsp. *lusitanica*	aerial parts	0.053–0.692	[48]
	*Lavandula viridis* L’Her	aerial parts	1.335–1.825	[11]
neochlorogenic acid (5-O-caffeoylquinic acid)	*Lavandula pedunculata* subsp. *lusitanica*	aerial parts	0.130–1.232	[48]
	*Lavandula viridis* L’Her	aerial parts	0.605–2.332	[11]
caftaric acid derivative	*Lavandula coronopifolia*	aerial parts	n.q.	[54]
chicoric acid (dicaffeoyltartaric acid)	*Lavandula coronopifolia*	aerial parts	n.q.	[54]
p-coumaric acid (4-hydroxycinnamic acid)	*Lavandula angustifolia*	herb	0.365–0.422	[58]
	*Lavandula angustifolia*	flowers	0.005	[1]
	*Lavandula spica*	herb	0.520	[57]
hydro-p-coumaric acid	*Lavandula spica*	herb	0.558	[57]
coumaric acid hexoside	*Lavandula dentata*	aerial parts	n.q	[5]
	*Lavandula stoechas*		n.q.	
ferulic acid (4-hydroxy-3-methoxycinnamic acid)	*Lavandula angustifolia*	herb	0.053–0.542	[58]
	*Lavandula angustifolia*	flowers	0.0002	[1]
	*Lavandula angustifolia* ‘Lady’	leaves	n.q.	[49]
	*Lavandula spica*	herb	0.380	[57]
	*Lavandula vera*	leaves	0.005	[52]
	*Lavandula viridis*	herb	n.q.	[68]
ferulic acid-4-*O*-glucoside	*Lavandula angustifolia* ‘Lady’	leaves	n.q.	[49]
lithospermic acid A	*Lavandula pedunculata*	flowering stems	n.q.	[16]
rosmarinic acid	*Lavandula angustifolia*	flowers	0.010	[1]
	*Lavandula coronopifolia*	aerial parts	n.q.	[54]
	*Lavandula pedunculata* subsp. *lusitanica*	aerial parts	0.011–6.246	[48]
	*Lavandula pedunculata*	flowering stems	n.q.	[16]
	*Lavandula dentata*	aerial parts	n.q.	[5]
	*Lavandula stoechas*	flowers	n.q.	[51]
	*Lavandula viridis* L’Her	aerial parts	1.346–20.714	[11]
salvianolic acid B (lithospermic acid B)	*Lavandula pedunculata*	flowering stems	n.q.	[16]
	*Lavandula stoechas*	aerial parts	n.q.	[5]
	*Lavandula stoechas*	flowers	n.q.	[51]
salvianolic acid C and G	*Lavandula coronopifolia*	aerial parts	n.q.	[54]
sinapic acid (4-hydroxy-3,5-dimethoxycinnamic acid)	*Lavandula angustifolia*	herb	0.362–2.352	[58]
yunnaneic acid F	*Lavandula dentata*	aerial parts	n.q.	[5]
	*Lavandula stoechas*		n.q.	
benzoic acid	*Lavandula spica*	herb	0.687	[57]
3-hydroxybenzoic acid	*Lavandula spica*	herb	0.018	[57]
4-hydroxybenzoic acid	*Lavandula angustifolia*	herb	0.002	[58]
	*Lavandula angustifolia*	flowers	0.011	[1]
	*Lavandula spica*	herb	1.578	[57]
	*Lavandula vera*	leaves	0.002	[52]
vanillic acid (4-hydroxy-3-methoxybenzoic acid)	*Lavandula angustifolia* *Lavandula angustifolia* *Lavandula vera*	herb flowers leaves	0.003–0.010 0.0007 0.001	[58] [1] [52]
syringic acid (4-hydroxy-3,5-dimethoxybenzoic acid)	*Lavandula angustifolia*	herb	0.017–0.025	[58]
protocatechuic acid (3,4-dihydroxybenzoic acid)	*Lavandula angustifolia*	flowers	0.003	[1]
	*Lavandula angustifolia*	herb	0.007–0.047	[58]
	*Lavandula spica*	herb	0.301 × 10^−3^	[57]
gallic acid (3,4,5-trihydroxybenzoic acid)	*Lavandula angustifolia*	herb	0.005–0.017	[58]
	*Lavandula angustifolia*	flowers	0.0001	[1]
homoprotocatechuic acid (3,4-dihydroxyphenylacetic acid)	*Lavandula spica*	herb	0.007	[57]
homovanillic acid (4-hydroxy-3-methoxyphenylacetic acid)	*Lavandula spica*	herb	0.065	[57]

n.q.—no quantification.

**Table 4 molecules-28-00256-t004:** Flavonoids in different species of lavender.

Flavonoids	Species	Part of Plant	Contents [μg/g]	Ref.
**Flavones**				
apigenin (4’,5,7-trihydroxyflavone)	*Lavandula angustifolia*	herb	n.q.	[68]
	*Lavandula angustifolia* ‘Rosea’		n.q.	
	*Lavandula stoechas*		n.q.	
	*Lavandula pedunculata* subsp. *lusitanica*	aerial parts	0.768–2.736	[48]
apigenin-O-glucoside	*Lavandula dentata*	aerial parts	n.q.	[5]
	*Lavandula stoechas*		n.q.	
apigenin-O-glucuronide	*Lavandula stoechas*	flowers	n.q.	[51]
apigenin hexoside	*Lavandula dentata*	aerial parts	n.q.	[5]
	*Lavandula stoechas*		n.q.	
genkwanin (7-methylapigenin)	*Lavandula dentata*	aerial parts	n.q.	[5]
	*Lavandula stoechas*		n.q.	
isoscutellarein-O-glucuronide	*Lavandula dentata*	aerial parts	n.q.	[5]
luteolin (3’,4’,5,7-terahydroxyflavone)	*Lavandula pedunculata* subsp. *lusitanica*	aerial parts	0.013–4.975	[48]
	*Lavandula viridis*	herb	n.q.	[68]
	*Lavandula viridis* L’Her	aerial parts	0.175–7.086	[11]
luteolin-O-glucoside	*Lavandula angustifolia*	herb	n.q.	[68]
	*Lavandula angustifolia* ‘Rosea’		n.q.	
	*Lavandula angustifolia* ‘Afropurpurea’		n.q.	
	*Lavandula lanata*		n.q.	
	*Lavandula stoechas*		n.q.	
	*Lavandula viridis*		n.q.	
	*Lavandula dentata*	aerial parts	n.q.	[5]
	*Lavandula stoechas*		n.q.	
	*Lavandula stoechas*	flowers	n.q.	[51]
luteolin-O-glucuronide	*Lavandula dentata*	aerial parts	n.q.	[5]
	*Lavandula stoechas*		n.q.	
	*Lavandula pedunculata*	flowering stems	n.q.	[16]
	*Lavandula stoechas*	flowers	n.q.	[51]
luteolin-O-hexosyl-O-glucuronide	*Lavandula pedunculata*	flowering stems	n.q.	[16]
methylluteolin-O-glucuronide	*Lavandula pedunculata*		n.q.	[16]
**Isoflavones**				
formononetin (7-hydroxy-4’-methoxyisoflavone)	*Lavandula spica*	herb	0.007	[57]
**Flavonols**				
quercetin (3,3’,4’,5,7-pentahydroxyflavone)	*Lavandula spica*	herb	0.016	[57]
quercetin 3-O-glucoside	*Lavandula stoechas*	flowers	n.q.	[51]
rutin (quercetin 3-rutinoside)	*Lavandula spica*	herb	0.283	[57]
taxifolin (dihydroquercetin)	*Lavandula spica*	herb	0.004	[57]
fisetin (5-desoxyquercetin)	*Lavandula spica*	herb	<0.001	[57]
myricetin (3,5,7,3’,4’,5’-hexahydroxyflavone)	*Lavandula angustifolia* “Rosea”	herb	n.q.	[68]
	*Lavandula lanata*		n.q.	
	*Lavandula viridis*		n.q.	
**Flavanol**				
(+)-catechin	*Lavandula vera*	leaves	0.004	[52]
**Flavanones**				
hesperetin (3’,5,7,-trihydroxy-4’-methoxyflavanone)	*Lavandula spica*	herb	0.001	[57]
hesperidin (hesperetin-7- rutinoside)	*Lavandula spica*	herb	0.023	[57]
neohesperidin (hesperetin 7-O-neohesperidoside)	*Lavandula spica*	herb	0.032	[57]
naringenin (4’,5,7-trihydroxyflavanone)	*Lavandula spica*	herb	0.398	[57]
	*Lavandula vera*	leaves	0.003	[52]
narirutin (naringenin 7-O-rutinoside)	*Lavandula spica*	herb	0.014	[57]
naringin (naringenin-7-neohesperidoside)	*Lavandula spica*	herb	0.001	[57]
eriodictyol (tetrahydroxyflavanone)	*Lavandula spica*	whole plant	0.007	[57]
eriodictyol-O-glucuronide	*Lavandula pedunculata*	flowering stems	n.q.	[16]
eriocitrin (eriodictyol 7-O-rutinoside)	*Lavandula spica*	herb	0.004	[57]
pinocembrin (dihydrochrysin)	*Lavandula spica*	herb	0.001	[57]
	*Lavandula viridis* L’Her	aerial parts	4.934–12.745	[11]
liquiritigenin (4’,7-dihydroxyflavanone)	*Lavandula spica*	herb	<0.001	[57]
liquiritin (7-hydroxyflavanone 4’-O-glucoside)	*Lavandula spica*	herb	0.002	[57]
vanillin (4-hydroxy-3-methoxybenzaldehyde)	*Lavandula angustifolia*	herb	0.100–0.193	[58]

n.q.—no quantification.

**Table 5 molecules-28-00256-t005:** Anti-inflammatory activity of *Lavandula* extracts.

*Lavandula* Species	Type of Extract	Animal Model of Inflammation	Anti-Inflammation Effect	Detection Method	Ref.
*Lavandula multifida*	ethanolic macerate, aqueous macerate	Croton-oil-induced ear edema in mice	edema reduction	HPLC	[7]
*Lavandula bipinnata*	Soxhlet extraction	-	inhibition of COX enzymes	HPTLC	[6]
*Lavandula officinalis*	hydroethanolic macerate	formalin test in mice	inhibition of COX enzymes	-	[93]
*Lavandula dentata*	hydromethanolic extracts (ASE)	carrageenan-induced paw edema in mice	decrease expression of iNOS, COX-2, IL-1β	RP-HPLC-DAD-MS	[5]
*Lavandula stoechas*			decrease expression of IL-1β, IL-6, TNF-α, iNOS, COX-2, MMP-9		
*Lavandula stoechas*	hydroethanolic extract (UAE)	carrageenan-induced paw edema in rats	edema reduction	-	[92]
*Lavandula pedunculata*	hydroalcoholic extracts	mouse macrophage-like cell line RAW 264.7 stimulated with LPS	inhibition of NO production	HPLC-DAD-ESI/MSn	[16]
	aqueous extracts (infusions)				

## Data Availability

Not applicable.

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
