# Peer review of "Phytochemical Profiling, Antioxidant and Anti-Inflammatory Activity of Plants Belonging to the Lavandula Genus"

_molecules, 2022, doi:10.3390/molecules28010256_

Round 1

Reviewer 1 Report

The article titled “Phytochemical profiling, antioxidant and anti-inflammatory activity of plants belonging to the Lavandula genus” is very well written, and provides information regarding ways to extract and the effects of other plants belonging to the Lavandula genus, however the article does not bring any image, inserting some graphs in place of some tables can bring the same information and make the article more interesting. To publish this article only a few adjustments, need to be made.

- Please merge the information from line 35 with line 37

-In extraction process section, please explain in the text if the hexane is the traditional extraction process at the industry

- Line 108, shorter extraction time for SFE? Please research about this, and adjust the sentence

- In terms of ultrasound extraction what is the best extraction frequency for this matrix?

-In terms of microwave assisted extraction, what is the best radiation range for this matrix?

- Line 156, please explain how the solvents are used in SFE? What is the maximum proportion of solvent that can be added and still maintain the supercritical state?

- Lines 262, 269, 286, 345: short sentences, please be careful with very short sentences and adjust them.

Author Response

Reviewer 1:

The article titled “Phytochemical profiling, antioxidant and anti-inflammatory activity of plants belonging to the Lavandula genus” is very well written, and provides information regarding ways to extract and the effects of other plants belonging to the Lavandula genus, however the article does not bring any image, inserting some graphs in place of some tables can bring the same information and make the article more interesting. To publish this article only a few adjustments, need to be made.

We are very grateful for your positive opinion and for your input to improve our manuscript.

To address your comment about making the article more interesting, the Table 1 was changed into Figure 1.

Figure 1. Global distribution map of various species belonging to the Lavandula genus. Inset: distribution of Lavandula genus in Europe.

  • Please merge the information from line 35 with line 37

The information from line 35 was merged with line 37.

Lavender infusions and lavender oil in the form of inhalation have sedative and anxiolytic effects which has been confirmed in both animal and human studies (9, 10, 12).

  • In extraction process section, please explain in the text if the hexane is the traditional extraction process at the industry

The above information about hexane was explained in the text.

It is a form of continuous hot extraction in which the target compounds are extracted from solids by repeated washing with organic solvents, such as ethyl acetate or hexane (24). This technique is often used in the industry, but it should be remembered that it is not environmentally friendly because it uses large amounts of toxic solvents. During the extraction of lavender essential oil with hexane, other substances such as waxes, pigments and albuminous materials are extracted in addition to volatile compounds. The hexane extracts obtained in this way can be purified, but this is time consuming and contributes to yield loss (4).

  • Line 108, shorter extraction time for SFE? Please research about this, and adjust the sentence

The changes were made to the manuscript.

These techniques are characterized by lower volumes of organic solvents (up to 100 mL), shorter extraction times (up to 1 hour, as compared with even days for e.g. maceration), and thus lower energy consumption (24).

Optimal extraction conditions to obtain the maximum yield and polyphenols content from lavender flowers were a temperature of 54.5 â—¦C, pressure of 29.79 MPa, and the extraction time of 45 min (47).

  • In terms of ultrasound extraction what is the best extraction frequency for this matrix?

The information was added to the manuscript.

The best extraction frequency for lavender extraction is 20-50 kHz (18, 32-34)

  • In terms of microwave assisted extraction, what is the best radiation range for this matrix?

The information was added to the manuscript.

Microwave power from 230 to 500 W is most often used to extract lavender (34, 37, 38)

  • Line 156, please explain how the solvents are used in SFE? What is the maximum proportion of solvent that can be added and still maintain the supercritical state?

The information was added to the manuscript.

The extraction procedure usually consist of static and dynamic phase. In the first phase the vessel with sample is filled with supercritical gas and thermostated for 10-15 minutes. During the dynamic phase the supercritical fluid is continuously passed through sample for about an hour and then the extract is collected in a receiver (4, 42). Solvents such as carbon dioxide, water, ethane, propane, and dimethyl ether could be used in supercritical extraction (43-44).

The addition up to 15-20 % of co-solvent (ethanol, methanol, water) to the supercritical fluid extraction process enables the extraction of polyphenolic compounds which are polar (46) . The total use of carbon dioxide during extraction depends on its pressure, flow rate, temperature,  sample size, and extraction time. In the study of Woźniak et al. 2017 (42) the CO2 consumption was 120-130 grams for a 10 gram sample and a CO2 flow of 1.8 g/min.

  • Lines 262, 269, 286, 345: short sentences, please be careful with very short sentences and adjust them.

The sentences were adjusted.

Line 262 (now lines 329-330): Flavonoids, occuring mainly as glycosides, are the most abundant group of polyphenols (73).

Line 269 (now lines 340-342): Among the isoflavones and flavanols present in lavender formononetin and catechin are occur most often, respectively (49, 57).

Line 286 (now lines 360-361): Free radicals contain at least one unpaired electron on the valence shell and react readily with the molecules in their vicinity, acting as prooxidants.

Line 345 (now lines 419-421): In  the presence of antioxidants, the radical cation is reduced, resulting in discoloration of the solution (58, 79), proportional to the antioxidant content in the sample (52).

Reviewer 2 Report

The article presents some interesting data about extracts obtained from Lavandula genus. 

However, before publication it requires some changes:

- please state clearly in the Introduction stated what is the novelty of this study

- for the extraction procedures presented: What are the gaps? Which method is best? What are the authors recommendations?   

- Rows 66-71 - please rephrase as all presented parameters are influencing the extraction

- Table 2 - the results of extraction (efficiency, yield) are more important to be presented and also the values obtained for antioxidant activity. Please add this information in the table.

- What are the specific characteristics of each species? What are the differences between species? What are the differences between the same specie grown in different area (country)? A detailed comparative analysis with support from the literature should be provided.

- Future recommendations are a critical part of the review. It should be provided.

Author Response

Reviewer 2: 

We would like to thank the Reviewer all the comments. We hope that in addressing them the manuscript was significantly improved.

The article presents some interesting data about extracts obtained from Lavandula genus.

However, before publication it requires some changes:

  • please state clearly in the Introduction stated what is the novelty of this study

The information was added to the manuscript.

Recent reviews on lavender concerned mainly the anti-anxiety, antidepressant and wound-healing properties of essential oil (9, 10, 12, 17, 18). However, to the best of our knowledge this work is the first attempt to review the data concerning the extraction and profile of other bioactive components of lavender, namely phenolic acids and flavonoids, as well as the antioxidant and anti-inflammatory properties of lavender phenolic-rich extracts.

  • for the extraction procedures presented: What are the gaps? Which method is best? What are the authors recommendations?

In an attempt to address this issues we have inserted the following text into the manuscript at the end of chapter 2:

As can be seen , in case of lavender the majority of studies used the traditional methods, i.e. maceration, infusion or digestion with water, alcohol or hydroalcoholic mixtures (17 studies). Among them the best results were obtained for maceration (11, 48), both in total yield and the content of phenolic compounds (Table 3). As for the modern techniques, the most often used (5 studies) was the ultrasound-assisted extraction. It is probably due to the relatively easy access to ultrasound devices in the scientific laboratories. From the analysis of various approaches to the extraction of bioactive compounds other than the essential oil components from lavender, the best method of extraction seems to be the ultrasound-assisted extraction with the hydroalcoholic solvent. As the phenolics in lavender are not very thermolabile, the elevated temperature (up to 60°C) could be used, increasing the yield. The use of ultrasound would decrease the extraction time compared with traditional methods.

  • Rows 66-71 - please rephrase as all presented parameters are influencing the extraction

The sentences were rephrased.

The extraction process is influenced by the physicochemical parameters of the solvents (boiling point, viscosity, density, vapor pressure and solvent power), their cost, non-flammability and non-toxicity (20, 24) as well as sample preparation (drying, grinding, sample particle size, and extraction parameters (extraction time, temperature, number of extraction steps, ratio of solvent to sample and use of co-solvent) (20, 22, 26, 27).

  • Table 2 - the results of extraction (efficiency, yield) are more important to be presented and also the values obtained for antioxidant activity. Please add this information in the table.

The information about the extraction yields and values obtained for antioxidant activity were added in the Table 2.

Additionally, in chapter 2 the following paragraph has been added:

Unfortunately, in many cases the yield of extraction cannot be determined because no dry extract was obtained (38, 49-52), and in some of the papers the mass of the extract and the extraction yield were not given.

Additionally, in chapter 4 the following paragraph has been added:

Unfortunately, the way in which the results of antioxidant activity are presented by different scientific teams is difficult to compare due to different conversions for extracts, different units and substances as equivalents. In addition, the extraction conditions such as different concentrations of reagents, solvents and temperature are responsible for different obtained values even for the same substance marked with the same test. We have made an effort to standardize these values, however, the publications lack many details required for such conversions.

  • What are the specific characteristics of each species? What are the differences between species? What are the differences between the same species grown in different area (country)? A detailed comparative analysis with support from the literature should be provided.

The specific characteristics of each species were presented in Table 1 (15, 16).

Additionally, in the Introduction the following paragraph has been added:

The species of the Lavandula genus presented in Table 1 differ in terms of height of the shrub, color of leaves, flowers and flowering period. Lavandula coronopifolia which occurs naturally in northern Africa  Arabian Peninsula and Western Asia, starts flowering the earliest, in January, whereas Lavandula pubescens, which occurs naturally in Syria, Jordan, Israel, Egypt, Yemen and Saudi Arabia, starts flowering from August to September. In recent years, more and more species have been cultivated outside their natural habitat. The same species grown in different areas may show morphological differences in the color of leaves, flowers, plant density or seed weight. The morphological variability, and thus the quantitative composition of individual compounds, is due to climatic conditions such as temperature, insolation, rainfall, humidity, altitude as well as environmental conditions such as fertilization, soil type and its pH level (16, 19).

The differences in the content of polyphenolic compounds between the same species grown in different area are presented in chapter 3 (Table 3 and Table 4).

  • Future recommendations are a critical part of the review. It should be provided.

Future recommendations were added to the Conclusions as the following text:

Each of the presented methods has its advantages and disadvantages. Which method a given laboratory chooses depends on many factors, such as the availability of equipment, solvents, trained staff, as well as maintenance costs.

However, since in the analyzed studies there was a great variation of both extraction protocols and plant material, there is a need of a systematic study comparing various extraction techniques of the same plant material to reliably recommend the optimal approach. The further studies are needed with the special attention paid to the optimization extraction and activity-guided extraction. Additionally, the key issue is to standardize the units and substances as equivalents in which the obtained results of activity determination are presented. There is also a gap concerning the extraction and analysis of lavender flower residues left after the distillation of essential oil – currently the main product of lavender – which could be a valuable source of phenolics exhibiting antioxidant and anti-inflammatory properties.

Round 2

Reviewer 2 Report

The article can be accepted.